# GFNet: A Deep Learning Framework for Breast Mass Detection

Xiang Yu [1,†], Ziquan Zhu [1,†], Yoav Alon [1], David S. Guttery [2] and Yudong Zhang [1,3,4,*]

1 School of Computing and Mathematical Sciences, University of Leicester, University Road, Leicester LE1 7RH, UK
2 Leicester Cancer Research Centre, University of Leicester, University Road, Leicester LE2 7LX, UK
3 School of Computer Science and Technology, Henan Polytechnic University, Jiaozuo 454000, China
4 Department of Information Systems, Faculty of Computing and Information Technology, King Abdulaziz University, Jeddah 21589, Saudi Arabia
* Correspondence: yudongzhang@ieee.org; Tel.: +44-754-870-0453
† These authors contributed equally to this work.

**Abstract:** Background: Breast mass is one of the main symptoms of breast cancer. Effective and accurate detection of breast masses at an early stage would be of great value for clinical breast cancer analysis. Methods: We developed a novel mass detection framework named GFNet. The GFNet is comprised of three modules, including patch extraction, feature extraction, and mass detection. The developed breast mass detection framework is of high robustness and generality that can be self-adapted to images collected by different imaging devices. The patch-based detection is deployed to improve performance. A novel feature extraction technique based on gradient field convergence features (GFCF) is proposed to enhance the information of breast mass and, therefore, provide useful information for the following patch extraction module. A novel false positives reduction method is designed by combining the texture and morphological features in breast mass patch. This is the first attempt at fusing morphological and texture features for breast mass false positive reduction. Results: Compared to other state-of-the-art methods, the proposed GFNet showed the best performance on CBIS-DDSM and INbreast with an accuracy of 0.90 at 2.91 false positive per image (FPI) and 0.99 at only 0.97 FPI, respectively. Conclusions: The GFNet is an effective tool for detecting breast mass.

**Keywords:** mass detection; gradient field convergence; deep learning

## 1. Introduction

Breast cancer is one of the most common cancers that threatens millions of people's lives around the world. Digital mammography (DM), which has proven to be a useful technique to reduce mortality [1], has been widely used as the standard imaging modality for breast cancer screening during the past few decades. However, the manual interpretation of mammograms can be time-consuming and sometimes challenging. Therefore, numerous computer-aided detection and diagnosis (CAD) systems have been developed to improve the efficiency and accuracy of mammography screening [2–4].

Breast mass, as one of the main symptoms of breast cancer, has received wide attention from the community. As a result, effective detection of breast mass allows radiologists to propose timely treatment. However, breast mass detection based on mammography itself is a challenging task, given the complexity of mammograms and the varied size of the mass. Usually, a full-field digital mammography (FFDM) image can be as large as 4000 by 3000 pixels, while the mass in presence can be as small as 100 by 100 pixels. Furthermore, the high density of breast tissue makes masses less distinguishable. Therefore, some research prefers to focus on the classification of annotated mass [5–7], which is the first step in developing efficient CAD systems [8–11]. However, developing automatic mass detection systems still has some limitations. First, the network they proposed may only be applicable to one kind of image modality. Second, the previous networks would detect the entire image, but the information that may be useful in the entire image is only concentrated

in a few patches. Third, the trained classifiers may not work well in test sets because of the complexity of mammograms. To this end, we developed a fully automated mass detection framework GFNet based on mammography images in this paper. Our system is a patch-based detection system that consists of three modules, including pre-processing, patch extraction, and mass detection. In the pre-processing stage, we mainly focused on obtaining the breast-only region from the mammograms and removing pectoral muscles, where we trained the Deeplabv3+ for the pectoral segmentation and removal task. Inspired by the work in [12], we proposed a novel patch extraction method for the following mass detection module in the patch extraction. Firstly, the first-order gradient field convergence feature (GFCF) of each pixel in the breast region is calculated. The gradient field convergence map (GFCM) is then formed by aggregating all GFCFs. Instead of thresholding GFCM by a fixed value, we proposed to binarize GFCFs into the binarized GFCM by keeping only the top ranked GFCFs. Then, the patches centered on the connected components in the binarized GFCM with fixed width are extracted as the interested regions. In the mass detection module, interested regions are firstly classified by deep-learning models, and false positive reduction, based on bagged decision trees, is applied to reduce the false positives. The deep-learning models are trained on ImageNet and are then transferred for breast mass and breast tissue classification. In false positive reduction, we proposed to extract morphological features from connected components in the binarized GFCM and texture features from the patches that survived the previous stage. The extracted features are fused to train bagged decision trees, while patches that survived are then taken as the detected masses. The contributions of this paper can be mainly concluded as follows:

1.  We proposed a novel patch-based mass detection framework, GFNet, that showed high performance on CBIS-DDSM and INbreast datasets. Moreover, the developed breast mass detection framework is of high robustness and generality that can be self-adapted to images collected by different imaging devices. Without adjustment, our proposed method showed consistent promising performance on CBIS-DDSM and INbreast datasets with an accuracy of 0.90 at 2.91 false positive per image (FPI) and 0.99 at only 0.97 FPI, respectively, which surpassed the state-of-the-art methods. The reason why we deploy patch-based detection is mainly two-fold. One is that local image patches containing mass candidates, instead of the entire image, are more preferable as a local feature of the mass is more precise. The second is that the proposed novel patch extraction method can effectively extract mass candidates from the whole image so that no global focus is required. Unlike other patch-based methods that simply divide the images into patches, the proposed method aimed at extracting only interested patches with a fixed-width from the images.

2.  In the proposed framework, we propose a novel feature extraction technique based on GFCF for mammograms in presence of breast mass. The proposed feature extraction technique is effective yet efficient and can enhance breast masses while suppressing other breast tissue. The success of the traditional sliding window-based breast mass detection methods relies on the computing devices to exhaustively extract patches from the mammograms without using the image information. Instead, the proposed feature extraction method selectively enhanced the information of breast mass and, therefore, provided useful information for the following patch extraction module. Moreover, the overall success of the proposed framework may contribute to the successful feature extraction method. In addition, the proposed feature extraction technique can be flexibly adapted to other application scenarios.

3.  In the proposed framework, we proposed a novel false positives reduction method. For patch-based breast mass detection frameworks, model training and testing procedures are quite different in that the trained classifiers may perform poorly on the testing set due to the complexity of mammograms. This makes false positive reduction a key yet challenging element in patch-based breast mass detection frameworks. Given the importance of the texture and morphological features in breast mass patch, we proposed to combine these two kinds of features for the reduction while multiple

machine-learning models are trained and the best one is used as the classifier. In addition, non-maximum suppression is then attached for further reduction. To our best knowledge, this is the first attempt at fusing morphological and texture features for breast mass false positive reduction.

The remainder of this paper is arranged as follows. In Section 2, we will briefly review related works concerning breast mass detection. In Section 3, we will introduce the details of our detection framework, followed by the experiments in Section 4. The details of the datasets involved in this research will be given in the experiments section. The discussion will be presented in Section 5, and we end this paper with a conclusion and suggestions for future work in Section 6.

## 2. Related Works

Generally speaking, breast mass detection methods can be divided into two categories, including one-stage and multi-stage methods. In the one-stage-based methods, unified detection networks such as YOLO are used for simultaneous detection and classification [13–15]. In the work [13], Al-antari et al. developed a simultaneous mass detection and classification system by introducing the YOLO detection framework. Six hundred original mammograms from the Digital Database for Screening Mammography (DDSM) are selected as the dataset. Those images are divided into five folds. Four of them are used for training, and the rest are used for validation. The trained YOLO-based system then detects the masses and has the detected masses classified into benign and malignant classes. The reported overall accuracy reached 99.7%. In another YOLO-based work [14], Ghada et al. effectively deployed the YOLO-V3 for the detection task, the latest version of the YOLO detection framework. On the public dataset named INbreast [16], the best result reported by the authors was 89.4% of detection, with an average precision of 94.2% and 84.6% for benign mass classification and malignant mass classification, respectively. An anchor-free one-stage network called BMassNet was developed in [15], where the authors proposed to use a dynamic updating training method for the training of the FSAF network [17]. On the INbreast dataset, the reported recall rate of each image was 0.930, while the FPI was 0.495.

Other mass detection works followed a routine way of detection in that regions of interest (ROIs) are first generated, followed by a classifier for classification [18–21]. The most straightforward way to generate ROIs for classification is to use the sliding window technique. In [18], a window with a fixed size of 224 by 224 pixels slides over mammograms at the stride of 56 by 56 pixels for candidate region extraction. Each generated patch is labeled as positive if the center pixels of masses are found in the patch. Otherwise, the patches are labeled as negative. Those patches are used to train deep-learning models to obtain the mass probability for each patch. A mass probability map (MPM) is then formed by aggregating all probabilities and binarized by a predefined threshold value. The final ROIs are then determined according to the bounding boxes of the components in the binarized MPM. The reported best model, which was based on InceptionV3 [22], showed a performance at a true positive rate (TPR) of $0.98 \pm 0.02$ at 1.67 FPIs on the INbreast dataset. Another patch-based method that proposes novel patch extraction and extraction methods was presented in [19]. In the proposed method, the density of the wavelet coefficient based on the Quincunx Lifting Scheme was used to generate mass region candidates. Based on the generated regions, a sliding window technique was then applied locally to extract the patches. Similar to other works, transfer learning was deployed for patch classification. On the INbreast dataset, the proposed method showed a detection accuracy of 0.98 with an FPI of 1.43. Two-stage detection frameworks, such as Faster-RCNN, are also an alternative path leading to the successful detection of mass [20]. Based on mammography images that are collected via a three-dimensional imaging modality called digital breast tomosynthesis (DBT), the proposed methods in [20] reported a sensitivity of 90% at 0.76 false positives per breast on a private dataset including 105 masses.

In this paper, we developed a novel mass detection framework called GFNet by integrating the proposed GFCF-based patch extraction method. A major problem with the

previous works based on detection frameworks is they worked best only when masses were obvious for detection. However, this can never be guaranteed in practice because of the availability of high-quality imaging instruments and the complexity of mammography images. Moreover, the detection performance of those methods can be further improved. Instead, our method showed high performance in breast mass detection on low-contrast mammograms and is friendly to transplantation.

## 3. Methodology

In this section, we will introduce the details of the three modules in our proposed GFNet framework, which are pre-processing, patch extraction, and mass detection.

### 3.1. Pre-Processing

Pre-processing mainly consists of pectoral removal and breast image enhancement. A mammogram is usually over 3000 pixels by 4000 pixels, while the breast region accounts for nearly 2/3 of the whole image. Within the breast region, the pectoral muscle is another distracting area that contributes to the increase in computational costs. Consequently, the computational cost can be effectively reduced when only the breast tissues in the breast region are considered. So, we believe it would be beneficial to the overall performance of the proposed framework if we have the pectoral muscle removed beforehand. Afterward, we applied image enhancement to improve the image contrast of mammograms for GFCF calculation.

Usually, mammograms have four views, including a left-side craniocaudal (LCC) view, a left mediolateral oblique (LMLO) view, and two counterparts from the right. The main difference between a CC-view mammogram and an MLO-view mammogram is that pectoral muscles are usually shown in MLO-view mammograms, while there is only a little or no pectoral muscle in CC-view mammograms. However, when implementing our framework, we found that the pectoral muscle should be removed as it affects the calculation of GFCF. To this end, we trained an improved version of Deeplabv3+ and completed the breast pectoral muscle removal task as in [23]. One segmentation example can be found in Figure 1. As can be seen from Figure 1b, the pectoral muscle has been successfully segmented and then removed in Figure 1c.

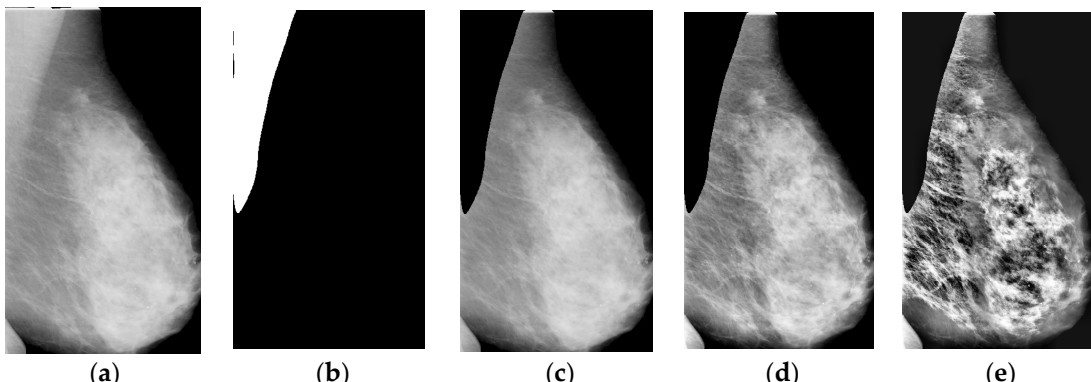

(a)  (b)  (c)  (d)  (e)

**Figure 1.** Pre-processing examples. (**a**) Breast-only image. We automatically extract the breast from a mammogram as we are only interested in the breast region. (**b**) Pectoral segmentation result. (**c**) Pectoral-muscle-removed breast image. Deeplab3+ model is trained for pectoral muscle segmentation. (**d**) Breast image enhanced by CLAHE when $\lambda$ is 0.05. (**e**) Breast image enhanced by CLAHE when $\lambda$ is 0.1.

After pectoral-muscle removal, the breast images are then enhanced by contrast-limited adaptive histogram equalization (CLAHE) due to the poor image contrast [24]. The enhanced images can be seen in Figure 1d, where $\lambda$ denotes the clip limit. Empirically, we set the value of $\lambda$ to be 0.05 as it provides the best visual image contrast improvement.

### 3.2. Patch Extraction

#### 3.2.1. Gradient Field Convergence Feature

The gradient field convergence feature (GFCF) is calculated for each pixel in the breast region. Given a pixel $A$ in the pre-processed breast image $I \in \mathbb{R}^{m \times n}$, $m$ and $n$ stand for the height and width of image $I$, respectively. A circular region $R$ with a radius of $r$ centering on $A$ has $N_A$ neighborhood pixels. Then, GFCF of the pixel $A$ can be defined as:

$$GFCF_A = \frac{1}{N_A} \sum_{j=1}^{N_A} (I_A - I_j) \tag{1}$$

where $I_A$ and $I_j$ stand for the intensity of pixels $A$ and $j$. By iteratively calculating GFCF for each pixel in the breast region of image $I$, we then have GFCM. However, we found that it was of high computational cost to go through all pixels in the breast region of the image. As a result, we introduced Algorithm 1 for the acquisition of GFCM.

---

**Algorithm 1**: Strided GFCF for GFCM

---

**Input**: Breast-only image $I \in \mathbb{R}^{m \times n}$, the total number of pixels $N_B$ in breast region $I_B$
**Expected output**: GFCM
**Step 1**: Calculate the mean intensity $M_R$
$M_R = \frac{1}{N_A} \sum_{j=1}^{N_A} I_B(j)$
**Step 2**: GFCM initialization
$GFCM \in \mathbb{R}^{m \times n}$ while being initialized as $-\infty$, which is a significantly large negative value.
**Step 3**: Calculation of GFCM
for $i = 1$ to $m$
  for $j = 1$ to $n$
    if $I_B(i, j) \geq M_R$
      $GFCM(i, j) = GFCF_{i,j}$
      else $GFCM(i, j) = -\infty$
where $GFCF_{i,j}$ is the GFCF at the location $(i, j)$.
**Step 4**: Normalization of GFCM
$GFCM(i, j) = 255 \times \frac{GFCM(i,j) - \min(GFCM)}{\max(GFCM) - \min(GFCM)}$
where $\max(.)$ and $\min(.)$ correspond to maximum and minimum operations, respectively.

---

#### 3.2.2. Patch Extraction

We then have GFCFs in the normalized GFCM ranked by their values, and only the top GFCFs are kept so that GFCM becomes GFCM'.

$$GFCM'(x) = \begin{cases} 1, if\ GFCF(x)\ in\ top\ k; \\ 0, if\ GFCF(x)\ not\ in\ top\ k; \end{cases} \tag{2}$$

To depress the noise speckles produced during the acquisition of GFCM', a morphological opening operation, which erodes the image first and then dilates the image, is deployed. Image patches regarding each connected component can then be extracted from the breast image with a fixed width accordingly. The resultant images corresponding to each procedure are shown in Figure 2.

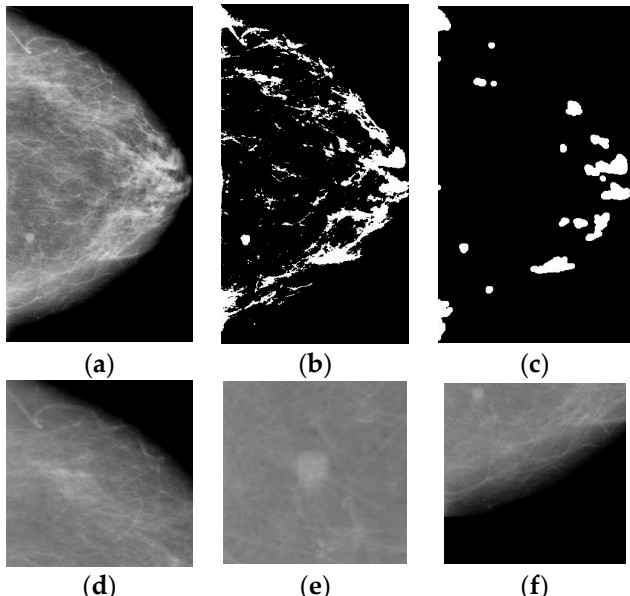

**Figure 2.** GFCM acquisition and patch extraction. (**a**) Enhanced breast image. (**b**) Binarized GFCM. (**c**) Binarized GFCM after the opening operation. (**d**) Extracted breast tissue patch. (**e**) Extracted breast mass patch that centers on the mass. (**f**) Extracted breast mass patch that contains mass.

### 3.3. Mass Detection

The mass detection module comprises two sub-modules, including patch classification and false positive reduction. Trickily, we converted the detection problem into a binary classification task. While there were still numerous false positives in the candidates, we then performed false positive reduction by machine-learning models trained via morphological and texture features. We then integrate non-maximum suppression (NMS) to reduce the false positives as it is usually involved in common object detection frameworks.

#### 3.3.1. Mass Detection

For patch classification, we transferred deep-learning models that were trained with images from ImageNet as classifiers in this study. Deep-learning models have shown a powerful performance on computer-vision tasks such as image classification and object detection [22,25–28]. Transfer learning has been found to be an efficient technique for obtaining deep-learning models with a promising performance at minimal training costs [29]. In this research, we implemented transfer learning by adapting the existing pre-trained state-of-the-art deep-learning models for the binary classification task. As the existing state-of-the-art models are developed for 1000 categories' classification, we, therefore, added two more fully connected layers with 256 and 2-dimensional output, respectively. We refer to the fully connected layer with X-dimensional output as FCX for simplicity here and after. To prevent the overfitting problem, the dropout technique is introduced by inserting a dropout layer between the layer FC1000 and layer FC256. The architectures of the adjusted models are shown in Figure 3. To train the deep-learning models, we applied the data augmentation technique to augment the mass patches, which are also called positive patches, in the training set, where the details will be shown later.

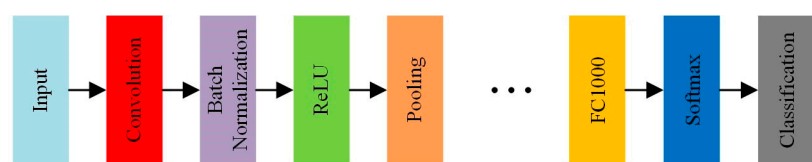

**Figure 3.** The architecture of the adjusted deep-learning models.

### 3.3.2. False Positive Reduction

After patch classification, we found that there were still many false positives. So, we then proposed to reduce false positives via machine-learning models and the NMS algorithm. The morphological features of the connected components and the texture features from corresponding patches are fused as new features that are forwarded to machine-learning models for feature learning. In the results of patch classification, we only keep $n$ top-scored patches instead and disregard the low-ranked patches. We then obtain texture features from those patches based on the gray-level co-occurrence matrix (GLCM), which is a statistical method that considers the spatial relationship of pixels. Four features, including contrast, correlation, energy, and homogeneity, are then derived from the obtained GLCM. The morphological features, including area, circularity, eccentricity, equivalent diameter, and solidity, are extracted from the connected components that correspond to the patches. We then fused these nine features by concatenating them, and the fused features are used to train multiple machine-learning models, including the supporting vector machine (SVM), bagged decision tree (BDT), and K-Nearest Neighbour (KNN) for the binary classification. By doing so, the patches that are wrongly classified as mass patches by the trained deep-learning model will be removed to the greatest extent. However, the bounding boxes of the detected masses may overlap with each other and, therefore, should be combined. Therefore, the detection results are then refined by applying the NMS algorithm. Details of the false positive reduction module can be found in Figure 4. The details of the proposed NMS algorithm can be seen in Algorithm 2.

---

**Algorithm 2**: Non-Maximum Suppression.

---

**Input**: Predicted scores: $Scores$ in $\mathbb{R}^n$, Locations: $location(cx, cy, width, height) \in \mathbb{R}^{4 \times n}$
**Expected output**: Refined scores: $Scores_R \in \mathbb{R}^{n'} (n' \leq n)$, Refined locations:
$location_R(cx, cy, width, height) \in \mathbb{R}^{4 \times n}$
**Step 1**: Set the $Scores$ **to be zero if the intersection** between them is over a predefined threshold value **Rate**.
for $i = 1$ to $n - 1$
   $Scores(j) = 0$;
     else $Scores(i) = 0$
where $IoU(i, j)$ denotes the area of intersection of union between $i$th and $j$th image patch, and the **Rate** is the intersection rate.
**Step 2**: Append the survived $Scores$ **and bounding box.**
$count = 1$
for $i = 1$ to $n$
   $Scores_R(count) = Scores(i)$
   $location_R(cx_{count}, cy_{count}, width_{count}, height_{count}) = location(cx_i, cy_i, width_i, height_i)$
   $count = count + 1$

---

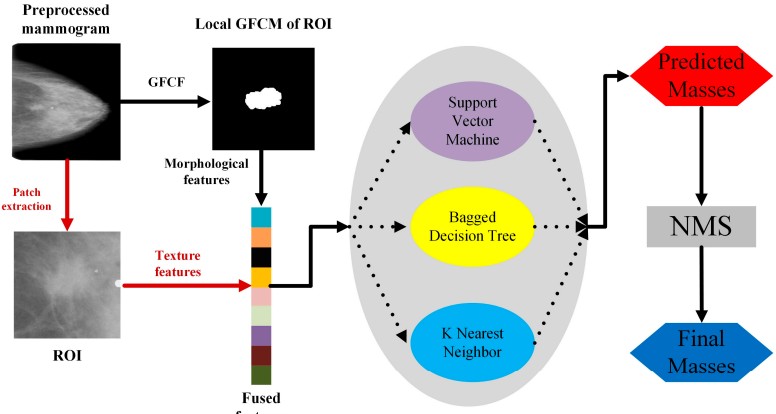

**Figure 4.** False positive reduction via machine-learning models.

### 3.4. Model Training and Inference

The model-training phase is a little bit different from the inference phase. In the training phase, the deep-learning models for patch classification and the classifiers for false positive reduction are two main components that can be trained with image patches. For the training of deep-learning models, the manually extracted patches are directly fed to the deep-learning models; for false positive reduction, the resultant patches, after patch extraction, are labeled and are taken as the inputs of the classifiers. The resultant patch is considered a positive patch if the intersection of the union (IoU) is great than 0.5. Otherwise, it is considered a negative patch. We skipped the evaluation of the proposed framework on entire images of the training set as it is likely the framework tends to show high performance. However, the overall detection performance of the proposed framework on the testing set relies on not only the classifiers with learnable parameters but also some predefined parameters such as the width of the extracted patches, the radius, *r*, of the circular region in patch extraction, and so forth. In addition, the overall detection performance on the testing set may vary slightly with or without some modules, such as the false positive reduction module or NMS. To explore the best choice of the non-learnable parameters and the combination of these modules, we will present detailed explorations in the model ablation section of the experiment. When calculating the detection results on the testing set, we reckon it is a successful detection of breast mass if the IoU between the detected mass and the true mass is greater than 0.2. The flowchart of the proposed method is given in Figure 5.

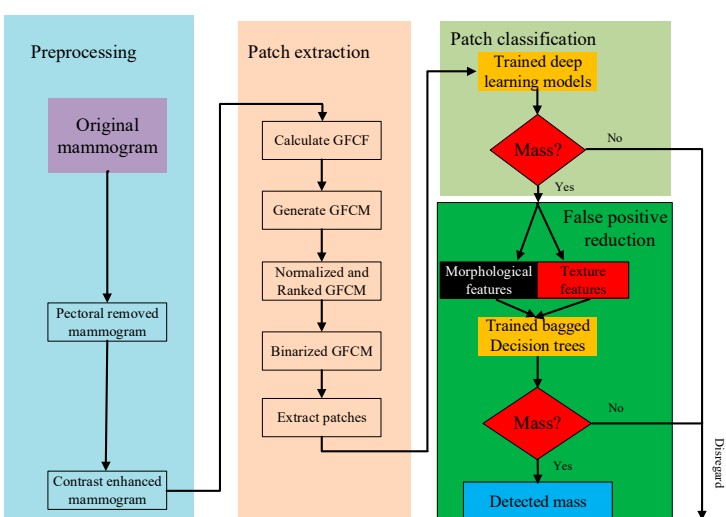

**Figure 5.** The flowchart of the proposed method.

## 4. Experiment

### 4.1. Datasets

In this study, we used two public datasets, including CBIS-DDSM and INbreast [16,30]. CBIS-DDSM, which is a carefully curated subset of the DDSM dataset, contains breast mass and breast calcification mammograms. Regarding these two categories, there are two sets of beforehand partitioned datasets with pixel-level annotated labels. Our focus is the subset of mass mammograms. However, we found some mammograms are not well annotated as the sizes between mammograms and the labels are different. So, we manually excluded those mammograms and labels, both in the training set and the testing set.

After the patch extraction procedure, we then obtained mass patches (or positive patches) and breast tissue patches (called negative patches). However, the number of negative samples greatly outnumbered that of positive samples. To form a balanced training set for patch classification, we augmented the positive samples by eight times and randomly selected the same number of patches from the negative samples. The eight

augmentation methods included flipping, rotation, contrast enhancement, etc. The details of the formed patch dataset are shown in Table 1. After the patch extraction module, there are, in total, 20,747 patches and 1911 of them are positive samples, while the remaining are negative samples.

**Table 1.** Information on the CBIS-DDSM dataset.

| Dataset | CC View | MLO View | Images in Total | Masses Patches | Negative Patches | Total |
|---|---|---|---|---|---|---|
| Training set | 541 | 625 | 1166 | 11,232 | 11,232 | 22,464 |
| Test set | 163 | 85 | 348 | 353 | 14,103 | 14,456 |

For the INbreast dataset, we used it for the overall performance evaluation of the proposed framework, and, therefore, no patches are extracted. While the INbreast dataset has 410 images in total, there are only 107 images containing masses, so we evaluated our framework on these 107 images instead.

*4.2. Classification Results by Deep-Learning Models*

In this study, we use *TP* for True Positive, *TN* for True Negative, *FP* for False Positive (*FP*), and *FN* for False Negative (*FN*). To quantify the performance of the deep-learning model for patch classification, we used various criteria including Area under the Curve (AUC) of the receiver operating characteristic curve, sensitivity, specificity, precision, $F1_{score}$, and accuracy. AUC is another important evaluation metric to measure the overall performance of the classification model. Sensitivity can be expressed by *TP* and *FN* as

$$Sensitivity = \frac{TP}{TP + FN} \tag{3}$$

Similarly, specificity can be denoted by *TN* and *FP* as

$$Specificity = \frac{TN}{TN + FP} \tag{4}$$

Precision, $F1_{score}$, and accuracy can then be written as

$$Precision = \frac{TP}{TP + FP} \tag{5}$$

$$F1_{score} = 2 \times \frac{Precision \times Sensitivity}{Precision + Sensitivity} \tag{6}$$

$$Accuracy = \frac{TP + TN}{TP + TN + FN + FP} \tag{7}$$

In this research, we explored multiple state-of-the-art deep-learning models for the binary classification task. The models are VGG19, ResNet50, ResNet101, InceptioinV3, DenseNet201, and InceptionResnetv2. For a better understanding of these models, we listed the details of the deep models from three perspectives, including the number of the training parameters, the number of layers in total, and the number of connections between the layers, as can be seen in Table 2.

We deployed the SPECTRE High-Performance Computing Facility at the University of Leicester for training as we were allowed access to a single GPU Tesla P100 PCI-E with a memory of 16 GB. The deep-learning framework is the deep-learning toolbox provided by Mathworks in Matlab 2019b. The parameters for training are shown in Table 3, where SGDM stands for Stochastic Gradient Descent with Momentum. The maximum training epoch is 9 in order to alleviate the overfitting problem. The initial learning rate is $10^{-3}$, which is the conventional setting. The minibatch size is 64, due to the size of the datasets

used in this paper. The learning rate drop period and learning rate drop rate are 3 and 0.3, which are set via experience. The optimization method is SGDM, which is a common choice. The shuffle of the train set is each epoch to improve the performance.

**Table 2.** Details of deep models.

| Name | Number of Training Parameters (Millions) | Number of Layers | Number of Connections |
|---|---|---|---|
| VGG19 | 143.7 | 50 | 49 |
| ResNet50 | 25.8 | 180 | 195 |
| ResNet101 | 44.8 | 350 | 382 |
| InceptionV3 | 24.1 | 318 | 352 |
| DenseNet201 | 20.2 | 711 | 808 |
| InceptionResNetv2 | 56.1 | 827 | 924 |

**Table 3.** Configurations of hyper-parameters.

| Parameters | Values |
|---|---|
| Maximum training epoch | 9 |
| Initial learning rate and | $10^{-3}$ |
| Minibatch size | 64 |
| Learning rate drop period | 3 |
| Learning rate drop rate | 0.3 |
| Optimization method | SGDM |
| Shuffle of the train set | Each epoch |

After training, we then tested the trained models with patches manually extracted from the testing set. To verify the effectiveness of data augmentation, we also evaluated the performance of the deep models that were only trained on the original training set, where the results can be seen in Table 2, while the ROCs can be seen in Figure 6. As can be seen from Table 4 and Figure 6, VGG19 turns out to be the best model as it provides the highest evaluation metrics and the AUC value. The reason behind this could attribute to the relatively large volume of VGG19 models, while the connection between the layers is straightforward. However, the precision and $F1_{score}$ are really low due to the biased distribution of the breast mass patches and breast background tissue patches.

**Table 4.** The performance of the deep-learning models on mass classification when trained on the original training set.

| Model | Sensitivity | Specificity | Precision | $F1_{score}$ | Accuracy |
|---|---|---|---|---|---|
| VGG19 | 0.88 | 0.91 | 0.19 | 0.32 | 0.91 |
| ResNet50 | 0.88 | 0.88 | 0.16 | 0.29 | 0.88 |
| ResNet101 | 0.82 | 0.88 | 0.15 | 0.25 | 0.88 |
| InceptionV3 | 0.86 | 0.89 | 0.16 | 0.27 | 0.8 |
| DenseNet201 | 0.86 | 0.88 | 0.15 | 0.28 | 0.88 |
| InceptionResNetv2 | 0.80 | 0.89 | 0.15 | 0.26 | 0.89 |

For comparison, we then measured the performance of deep-learning models trained on the augmented training set in terms of the mentioned metrics and listed the results in Table 5. The receiver operating characteristic (ROC) curves and AUC are given in Figure 7, where VGG19 turns out to be the best-performed classifier as it obtained the biggest AUC. As can be seen, InceptionV3 and InceptionResNetv2 performed best in terms of overall accuracy. However, it is likely these two deep models are suffering from overfitting as they obtained low sensitivity. Note that the testing dataset is biased, while the number of tissue background patches greatly outnumbered that of the mass patches. As a result, the

AUC and the sensitivity are more important compared to other evaluation metrics. Instead, VGG19 has the highest sensitivity, though the overall accuracy is lower compared to that of the other models. Considering the overall performance of different models, we only use VGG19 as the deep-learning model for breast mass patch classification.

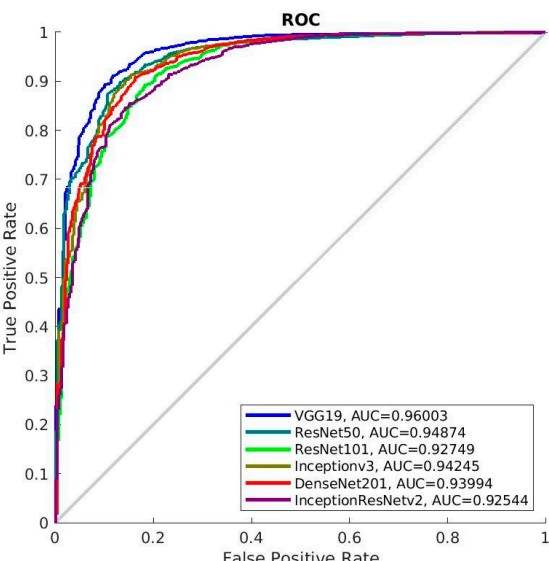

**Figure 6.** ROCs of the deep-learning models that are trained on the original training set.

**Table 5.** The performance of the deep-learning models on mass classification when trained on the augmented training set.

| Model | Sensitivity | Specificity | Precision | $F1_{score}$ | Accuracy |
|---|---|---|---|---|---|
| VGG19 | 0.90 | 0.95 | 0.29 | 0.44 | 0.94 |
| ResNet50 | 0.82 | 0.95 | 0.28 | 0.42 | 0.94 |
| ResNet101 | 0.81 | 0.96 | 0.34 | 0.48 | 0.96 |
| InceptionV3 | 0.65 | 0.98 | 0.48 | 0.27 | 0.98 |
| DenseNet201 | 0.71 | 0.96 | 0.30 | 0.43 | 0.95 |
| InceptionResNetv2 | 0.61 | 0.98 | 0.48 | 0.54 | 0.97 |

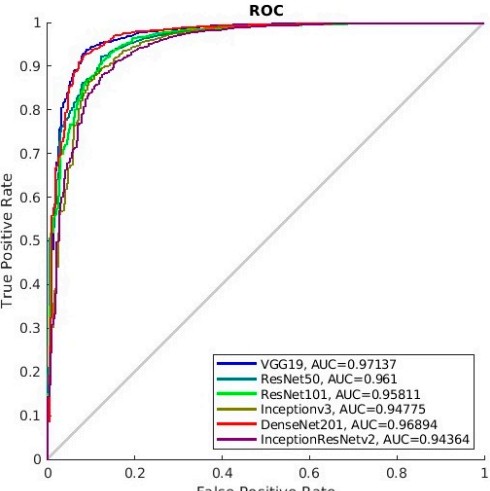

**Figure 7.** ROCs of the deep-learning models that are trained on the augmented training set.

As aforementioned, for the classifiers for false positive reduction, we chose classifiers including SVM, BDT, and KNN. We used five cross-validations to evaluate the performance

of these models while we partitioned the positive patches and negative patches evenly into each fold. We listed the results of the trained models in Table 6, where $KNN_x$ stands for the KNN model with $x$ as the number of neighbors. Similarly, $y$ in $BDT_y$ stands for the number of the decision trees in the BDT model. For evaluation metrics, we simply take the three most representative metrics, including specificity, sensitivity, and accuracy, for consideration. As can be seen from the table, $KNN_{20}$ showed highest averaged performance regarding the classification task, and we then took the $KNN_{20}$ model that performed best amongst the trained models as the classifier for false positive reduction. While the average sensitivity of the trained $KNN_{20}$ models is low, the $KNN_{20}$ model with best performance, however, showed over 90% of sensitivity and is assumed to be suitable to be a false positive reduction model.

**Table 6.** Performance of the trained classifiers on the generated patches of the training set (%).

| Model | Sensitivity | Specificity | Accuracy |
|---|---|---|---|
| SVM | $91.06 \pm 1.52$ | $13.12 \pm 5.78$ | $70.73 \pm 15.70$ |
| $KNN_5$ | $91.01 \pm 0.05$ | $26.29 \pm 4.44$ | $90.18 \pm 0.15$ |
| $KNN_{10}$ | $90.91 \pm 0.05$ | $55.98 \pm 13.35$ | $90.82 \pm 0.05$ |
| $KNN_{20}$ | $90.85 \pm 0.04$ | $58.00 \pm 37.68$ | $90.83 \pm 0.06$ |
| $BDT_5$ | $91.16 \pm 0.07$ | $23.32 \pm 2.36$ | $89.43 \pm 0.17$ |
| $BDT_{10}$ | $91.09 \pm 0.07$ | $40.59 \pm 8.14$ | $90.60 \pm 0.17$ |
| $BDT_{20}$ | $91.05 \pm 0.07$ | $38.51 \pm 6.45$ | $90.59 \pm 0.10$ |

*4.3. Model Ablation on CBIS-DDSM Dataset*

In this section, we will explore the key components that will lead to higher performance of the developed system. The detection capability of the system mainly relies on the performance of the trained deep-learning models, where VGG19 is chosen. The number of FPI generated by the system relies on the performance of the trained classifiers and NMS module, where the importance of the trained classifiers and NMS module needs to be verified via experiments. Furthermore, the overall detection performance of the proposed framework also relies on the following parameters:

1.  The size of extracted patches, which can be denoted as *width* determines the scope of each patch and, therefore, directly determines the scale of the input; if the *width* is too small, the mass may not be totally included in the patch, and the performance of the deep-learning model may also be affected. A larger *width* of patches, however, means further bounding-box-refinement procedures are required as the mass can be found in a larger area. Consequently, we also need to explore the relationship between the overall performance of the proposed framework and the size for patch extraction.

2.  The radius, *r*, in the calculation of GFCF. *r*, which can be interpreted as the size of filters in convolution, determines the number of neighborhood pixels that will be involved in the calculation of GFCF. A smaller *r* may focus on small object-like features, while a larger *r* increases the computational cost by a second order because the number of pixels for GFCF calculation is proportional to $r^2$. Finally, we must specify an optimal *r* for the trade-off between computational cost and the receptive field of GFCF.

3.  *k*, which determines the proportion of top GFCFs to be kept in GFCM'; a smaller *k* may lead to a few numbers of connected components in GFCM'. As a result, some ROI regions could be missing when extracting ROI patches. However, if *k* is too large, then connected components in GFCM' may connect to each other and result in larger connected components, which makes it challenging to choose the proper patch width. So, we will explore the optimal value of *k* that gives the best results on the testing set of CBIS-DDSM.

4.  *n*, the number of patches with top scores. When testing the trained models, it is likely that these models performed poorly on the testing set. So, we chose the top *n* patches to keep as many mass patches as possible in the beginning. Another reason why we did not choose a fixed threshold value is that the predictive scores for patches

generated through our proposed patch extraction method may fall below the threshold value as the breast tissues are complicated. However, the choice of $n$ should be careful as a larger $n$ tends to increase FPI, while a smaller $n$ tends to reduce the detection accuracy. The impact of $n$ will be shown in the experimental part.

We first checked the performance of the patch extraction module via GFCM under different configurations of these parameters and listed both patch extraction results and detection results in Table 7.

**Table 7.** Performance of patch extraction and mass detection on CBIS-DDSM with NMS.

| *Width* | *r* | *k* | *n* | **Successful Patch Extraction Rate** | *Accuracy@FPI* |
|---------|-----|-----|-----|--------------------------------------|----------------|
| 129 | 30 | 0.08 | 15 | 0.88 | 0.75@3.20 |
| 149 | 30 | 0.08 | 15 | 0.91 | 0.79@2.75 |
| 169 | 30 | 0.08 | 15 | 0.91 | 0.82@2.52 |
| 199 | 30 | 0.08 | 15 | 0.93 | 0.82@2.13 |
| 129 | 60 | 0.15 | 15 | 1.00 | 0.57@4.14 |
| 149 | 60 | 0.15 | 15 | 1.00 | 0.68@3.66 |
| 169 | 60 | 0.15 | 15 | 0.99 | 0.74@3.39 |
| 199 | 60 | 0.15 | 15 | 0.99 | 0.83@2.99 |
| 129 | 60 | 0.15 | 20 | 1.00 | 0.57@4.32 |
| 149 | 60 | 0.15 | 20 | 1.00 | 0.68@3.83 |
| 169 | 60 | 0.15 | 20 | 0.99 | 0.74@3.55 |
| 199 | 60 | 0.15 | 20 | 0.99 | 0.83@3.16 |

The value of **_Rate_** in the NMS algorithm is 0.5 by default. We consider it a successful extraction of the patches if the mass patches have been included in all the patches generated from a mammogram. Specifically, for the detection results here, we simply deployed the NMS algorithm without applying the false positive reduction as we aimed at preparing coarse detection results for false positive reduction. The accuracy in the table here and after indicates the rate between the number of detected numbers and the number of real breast masses, while the FPI is the average number of false positives in each image.

To begin with, we set the $r$, $k$, and $n$ to be 30, 0.08, and 15, respectively, while varying *width* is from 129 to 199. As can be seen, the patch extraction rate increases along with the *width*, which means a larger *width* contributes to better performance of patch extraction. In addition, the detection performance benefited from the *width* as high accuracy, and lower FPI is seen when the *width* increases. We then increased $r$, $k$, and $n$ to 60, 0.15, and 15, respectively. As can be seen, the patch extraction rate reaches 1, which means the module indeed benefited from the increase of these parameters. However, the overall detection performance becomes even worse because the FPI increased significantly while the overall accuracy decreased slightly. The reason behind this is that the larger values of $r$, $k$, and $n$ lead to more generated patches. We then increase the $n$ to 20 to evaluate the impact of the $n$ on the detection performance. As a result, the FPI increases while leaving the same accuracy. Therefore, $n$ should be no more than 15. Nevertheless, we will explore more about the parameter setting in the later experiment.

We then tested the detection system with a false positive module but without no NMS. Given the fact that $KNN_{20}$ performed best among all classifiers, we then took the trained $KNN_{20}$ with the best performance as the classifier. The detection results can be seen in Table 8.

As can be seen from the above table, the overall performance of the detection system has been greatly improved thanks to the false positive reduction module. By introducing the module, more breast mass patches have been correctly recognized while more breast tissue patches are wrongly recognized as a breast mass, which leads to the increase of FPI. When we fixed $r$, $k$, and $n$ to be 60, 0.15, and 20, we found that a larger *width* tends to contribute to higher detection accuracy. We then fixed the *width*, $k$, and $n$ to be 149, 0.15, and 20 but varied $r$ from 40 to 60. The accuracy remains unchanged, but a small $r$ leads to a higher FPI. Considering this, a large $r$ is preferable. However, a too large $r$ will increase

the overall computational cost. Therefore, we choose $r = 60$ in the later experiment. When varying $k$ and fixing other parameters, we found that the connected components tend to connect when $k$ is larger than 0.15 because more pixels are introduced. So, we only tested the varied values, not beyond 0.15. From the table above, the accuracy and the FPI seem to increase when $k$ increases. We finally tested the model with different values of $n$, and the conclusion is that FPI increases when $n$ increases as more patches are considered in the mass patch classification stage. The accuracy, however, only reaches the highest when optimal $n$ is chosen. Based on previous experiments, we then set $r$, $k$, and $n$ to be 60, 0.15, and 15 while varying the *width* from 149 to 199 in a later experiment. The reason why we vary the *width* is that a smaller *width* tends to provide more accurate location information of the detected masses.

**Table 8.** Performance of the proposed GFNet framework on CBIS-DDSM parameters with $KNN_{20}$ as the classifier of false positive reduction and without NMS.

| *Width* | *r* | *k* | *n* | *Accuracy@FPI* |
|---------|-----|-----|-----|----------------|
| 129 | 60 | 0.15 | 20 | 0.83@4.17 |
| 149 | 60 | 0.05 | 20 | 0.79@3.23 |
| 149 | 60 | 0.10 | 20 | 0.85@3.83 |
| 149 | 60 | 0.15 | 20 | 0.86@4.35 |
| 149 | 60 | 0.15 | 5 | 0.82@2.20 |
| 149 | 60 | 0.15 | 10 | 0.86@3.28 |
| 149 | 60 | 0.15 | 15 | 0.86@3.69 |
| 149 | 60 | 0.15 | 20 | 0.86@3.94 |
| 169 | 60 | 0.15 | 20 | 0.87@3.81 |
| 199 | 60 | 0.15 | 20 | 0.90@3.82 |

Finally, we tested the detection model with trained $KNN_{20}$ and NMS. The detection results can be seen in Table 9. Note that **Rate** is the only parameter that should be determined for the NMS algorithm, and we then vary it from 0.5 to 0.7. As can be seen, the FPI under different situations has been greatly reduced compared to the models without NMS. Moreover, the models with trained $KNN_{20}$ and NMS showed higher detection accuracy compared to the accuracy of the models with only NMS, while they have close values of FPI. Therefore, we believe the previous experiments showed the effectiveness of each module in the developed system. Some detection results can be seen in Figure 8. Figure 8a–d are successful detection examples, where we can see that the breast masses have been detected without FPIs. One deficiency of the proposed method is the failure of detection of breast mass within the pectoral muscle, as can be seen in Figure 8a. Furthermore, the complexity of the breast tissues, especially the breast mass-like tissue, posed a great threat to the detection of the proposed method, which can be seen from Figure 8f–h.

**Table 9.** Performance of the proposed GFNet framework on CBIS-DDSM parameters with $KNN_{20}$ as the classifier of false positive reduction and NMS.

| *Width* | *r* | *k* | *n* | *Rate* | *Accuracy@FPI* |
|---------|-----|-----|-----|--------|----------------|
| 149 | 60 | 0.15 | 15 | 0.5 | 0.86@2.97 |
| 169 | 60 | 0.15 | 15 | 0.5 | 0.87@2.64 |
| 199 | 60 | 0.15 | 15 | 0.5 | 0.89@2.21 |
| 149 | 60 | 0.15 | 15 | 0.6 | 0.86@3.22 |
| 169 | 60 | 0.15 | 15 | 0.6 | 0.87@2.94 |
| 199 | 60 | 0.15 | 15 | 0.6 | 0.89@2.60 |
| 149 | 60 | 0.15 | 15 | 0.7 | 0.86@3.44 |
| 169 | 60 | 0.15 | 15 | 0.7 | 0.87@3.18 |
| 199 | 60 | 0.15 | 15 | 0.7 | 0.90@2.91 |

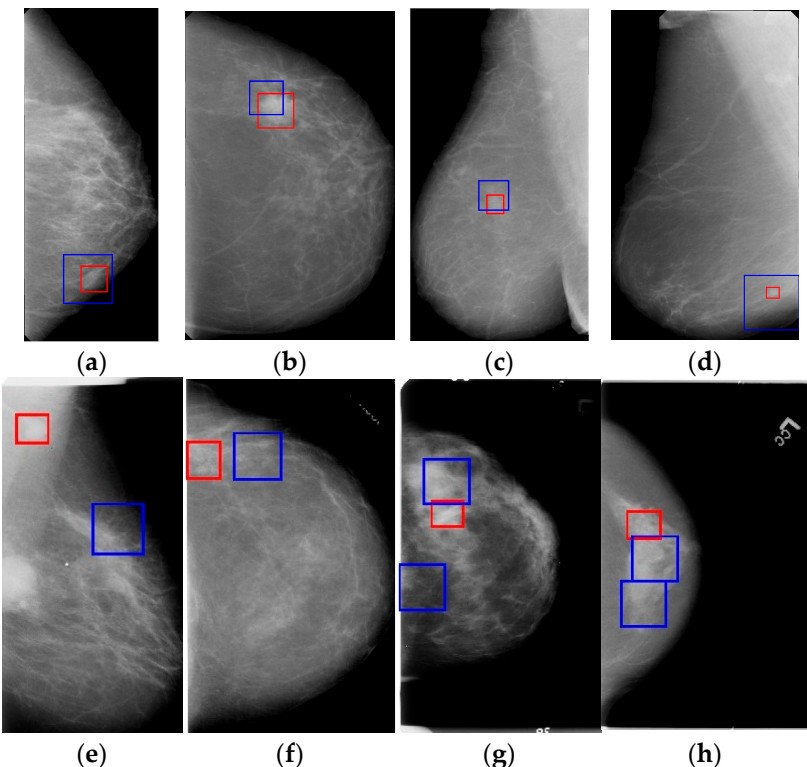

**Figure 8.** Detection results on CBIS-DDSM by the proposed detection framework. The red rectangles indicate the ground truth of masses while the blue rectangles stand for the detection results. (**a**) Detected mass: 1, true mass: 1, FPI: 0. (**b**) Detected mass: 1, true mass: 1, FPI: 0. (**c**) Detected mass: 1, true detected mass: 1, FPI: 0. (**d**) Detected mass: 1, true mass: 0, FPI: 1. (**e**) Detected mass: 1, true mass: 0, FPI: 1. (**f**) Detected mass: 2, true mass: 0, FPI: 2. (**g**) Detected mass: 2, true mass: 0, FPI: 2. (**h**) Detected mass: 2, true mass: 0, FPI: 2.

### 4.4. Detection Results on INbreast

Without any adaptation of the proposed framework, we directly evaluated our framework on 107 mammograms from the INbreast dataset that contains at least one mass per mammogram. We fixed the *width*, *r*, and *k*, but varied *n*, and the results based on the models without false positive reduction but with NMS are shown in Table 10. By default, we set the **Rate** in the NMS algorithm to 0.5.

**Table 10.** Performance of the proposed GFNet framework on INbreast without false positive reduction but with NMS.

| Width | r | k | n | Accuracy@FPI |
|-------|-----|------|-----|--------------|
| 149 | 60 | 0.15 | 5 | 0.95@0.72 |
| 149 | 60 | 0.15 | 10 | 0.97@1.04 |
| 149 | 60 | 0.15 | 15 | 0.97@1.34 |
| 149 | 60 | 0.15 | 20 | 0.97@1.56 |

As can be seen from Table 10, the overall performance of the proposed framework reaches the best one when the *width* is 149, *r* is 60, *k* is 0.15, and *n* is 10. Especially, the FPI is much lower compared to the results on the testing set of CBIS-DDSM. The reason could be that the resolutions of mammograms from the INbreast dataset are higher than that of the mammograms from CBIS-DDSM. With the increase of *n*, the accuracy seems to be saturated while the FPI increases, which indicates that *n* should be carefully chosen instead of setting it as large as possible. Additionally, the results further show the robustness and effectiveness of our GFNet framework. Note that deep-learning models and the bagged

decision trees were only trained with the training set of CBIS-DDSM while no further fine-tuning process is applied to INbreast.

We then examine the performance of GFNet with false positive reduction via trained $KNN_{20}$ but without NMS on INbreast, and the detection results can be seen in Table 11. As can be seen, the attached false positive reduction module helps to boost the detection performance. However, the FPI also increases compared to the FPI produced by the previous models with only NMS, which shows the effectiveness of the NMS module. The best model obtained is the one with the *width* = 149, *r* = 60, *k* = 0.15, and *n* = 10 which achieved an accuracy of 0.99 at 1.37 FPI. However, if we look at the lowest value of FPI but with acceptable accuracy, the model with the *width* = 149, *r* = 60, *k* = 0.15, and *n* = 5 becomes the best one.

**Table 11.** Performance of the proposed GFNet framework on INbreast with false positive reduction but without NMS.

| Width | r | k | n | Accuracy@FPI |
|-------|-----|------|-----|--------------|
| 149 | 60 | 0.15 | 5 | 0.97@0.81 |
| 149 | 60 | 0.15 | 10 | 0.99@1.37 |
| 149 | 60 | 0.15 | 15 | 0.99@1.89 |
| 149 | 60 | 0.15 | 20 | 0.99@2.33 |

Finally, we tested the full version of GFNet with false positives and NMS on INbreast and concluded the results in Table 12. Based on previous experiments, we found that the detection accuracy on INbreast is much higher than that of the accuracy on CBIS-DDSM. We, therefore, propose to vary *n* from 5 to 15. To evaluate the values of **Rate** to the overall detection performance on INbreast, we varied the **Rate** from 0.5 to 0.7. As can be seen from Table 11, the overall detection accuracy increases along with the increase in the **Rate**. However, the FPI increases as well when the **Rate** increases. Finally, the model with the *width* = 149, *r* = 60, *k* = 0.15, *n* = 5, and *Rate* = 0.5 gives the lowest FPI, while the model with the *width* = 149, *r* = 60, *k* = 0.15, *n* = 10, and *Rate* = 0.5 gives the highest detection accuracy of 0.99 at FPI of 0.97.

**Table 12.** Performance of the proposed GFNet framework on INbreast with false positive reduction and NMS.

| Width | r | k | n | Rate | Accuracy@FPI |
|-------|-----|------|-----|------|--------------|
| 149 | 60 | 0.15 | 5 | 0.5 | 0.97@0.65 |
| 149 | 60 | 0.15 | 10 | 0.5 | 0.99@0.97 |
| 149 | 60 | 0.15 | 15 | 0.5 | 0.99@1.28 |
| 149 | 60 | 0.15 | 5 | 0.6 | 0.97@0.70 |
| 149 | 60 | 0.15 | 10 | 0.6 | 0.99@1.08 |
| 149 | 60 | 0.15 | 15 | 0.6 | 0.99@1.46 |
| 149 | 60 | 0.15 | 5 | 0.7 | 0.97@0.74 |
| 149 | 60 | 0.15 | 10 | 0.7 | 0.99@1.20 |
| 149 | 60 | 0.15 | 15 | 0.7 | 0.99@1.52 |

Some detection examples can be seen in Figure 9, where true breast masses are detected at a lower cost of FPIs. Figure 9a–d are successful detection results without FPIs, which shows the effectiveness of the proposed detection method. Figure 9e–h are detection results with FPIs. The mass in the pectoral muscle is removed along with the removal of the pectoral muscle, and that is why the image in Figure 9e will have multiple FPIs but no truly detected masses. Besides the true masses in the mammograms, there are also mass-like tissues that confuse the detection system, which we can tell from Figure 9f–h.

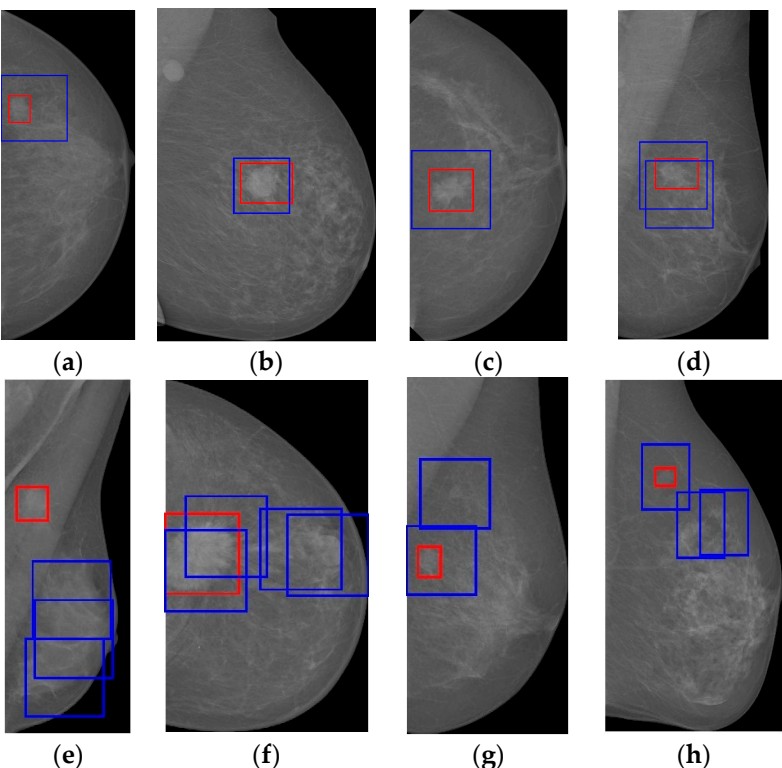

**Figure 9.** Detection examples on INbreast by VGG19 with false positive reduction and NMS. The red rectangles indicate the ground truth of masses while the blue rectangles stand for the detection results. (**a**) Detected mass: 1, true mass: 1, FPI: 0. (**b**) Detected mass: 1, true mass: 1, FPI: 0. (**c**) Detected mass: 1, true detected mass: 1, FPI: 0. (**d**) Detected masses: 3, true detected mass: 0, FPI: 3. (**e**) Detected mass: 4, true mass: 1, FPI: 2. (**f**) Detected mass: 2, true mass: 1, FPI: 1. (**h**) Detected mass: 3, true mass: 1, FPI: 2.

### 4.5. Method Comparison

We then compared our method with the state-of-the-art methods and listed the results in Table 13. As can be seen, our methods showed the best performance among all the methods, which further supported the effectiveness of the proposed framework.

**Table 13.** Method comparison.

| Method | Dataset | Number of Images for Evaluation | Performance |
| --- | --- | --- | --- |
| de Sampaio, et al. [31] | DDSM | 70 | 0.84@0.19 |
| Diniz, et al. [32] | DDSM | 54 | 0.90@0.88 |
| Andreadis, et al. [33] | CBIS-DDSM | 73 | 0.81@1.62 |
| Our method | CBIS-DDSM | 348 | 0.89@2.21 |
| Our method | CBIS-DDSM | 348 | 0.90@2.91 |
| Hassan, et al. [34] | INbreast | 75 | 0.94@0.67 |
| Kozegar, et al. [35] | INbreast | 107 | 0.87@3.67 |
| Cao, et al. [15] | INbreast | 107 | 0.93@0.50 |
| Shen, et al. [36] | INbreast | 32 | 0.88@0.50 |
| NiroomandFam, et al. [19] | INbreast | 82 | 0.98@1.43 |
| Our method | INbreast | 107 | 0.97@0.65 |
| Our method | INbreast | 107 | 0.99@0.97 |

## 5. Discussion

There are some interesting issues we would like to discuss here. One is the necessity of pectoral segmentation. We implemented the GFCF extraction on mammograms without

pectoral removal. However, we found the salient edges of the pectoral will remain in the generated GFCM, which brings in more patches to be processed in the framework, while these patches are not quite helpful to the detection. So, we decided to remove the pectoral muscle from the mammograms. We also tried to keep only the patches with predicted scores beyond 0.5. However, one issue brought by this is that there could be no mass candidate patches that survive the first stage. Another reason why a fixed threshold value is inappropriate is because of the underperformance of the classification module on the testing set. Note that patches extracted from the training dataset are extracted regarding the true bounding boxes and, therefore, are more ideal than practical patches generated by our patch extraction method in the testing set. Another issue is the choice of the *Rate* in the NMS algorithm. A low value of the *Rate* would lead to easier suppression and, therefore, contribute to a lower FPI. However, the detection accuracy will decrease along with the decrease in FPI.

## 6. Conclusions

In this work, we proposed a novel patch-based mass detection framework GFNet, which showed promising results on two public datasets. Compared to traditional patch-based mass detection methods, we proposed a novel and effective patch extraction method by introducing a ranked gradient field convergence feature. By introducing a two-stage stacked classification method, our proposed framework showed high performance of breast mass detection in both CBIS-DDSM and INbreast datasets at the cost of low FPIs, which surpassed the performance of the state-of-the-art methods. The novelty of this manuscript not only lies in the high performance of the framework but also in the novel feature extraction and false positive reduction techniques proposed in the framework.

Moreover, we provided a new breast mass detection strategy for mammography images. However, there are still some limitations to this work. Firstly, the performance of the classifiers, including deep-learning models, can be improved. Due to factors such as the limitation of the datasets, we only explored limited numbers of the deep-learning models and the machine-learning models. More work can be done to contribute to classifiers with a higher performance and, therefore, lead to a higher detection performance with the existing framework. The second limitation is that only patch-level detection results can be obtained by our detection framework. While a patch-level detection result is enough for further classification tasks such as breast cancer classification, more work can be conducted to retrieve more accurate detection results via the connected components in GFCM. Nevertheless, the newly proposed detection framework, GFNet, has a powerful capability of feature extraction and overall architecture. As a result, we hope this work could provide future works with some meaningful inspiration.

**Author Contributions:** X.Y.: Conceptualization; software; data curation; writing—original draft; writing—review and editing; visualization. Z.Z.: methodology; software, validation; investigation; resources; writing—review and editing. Y.A.: methodology; software, validation; D.S.G.: investigation; resources; writing—review and editing; Y.Z.: methodology; formal analysis; investigation; data curation; writing—original draft; writing—review and editing; supervision; project administration; funding acquisition. All authors have read and agreed to the published version of the manuscript.

**Funding:** This paper is partially supported by MRC, UK (MC_PC_17171); Royal Society, UK (RP202G0230); BHF, UK (AA/18/3/34220); the Hope Foundation for Cancer Research, UK (RM60G0680); GCRF, UK (P202PF11); Sino-UK Industrial Fund, UK (RP202G0289); LIAS, UK (P202ED10, P202RE969); the Data Science Enhancement Fund, UK (P202RE237); the Fight for Sight, UK (24NN201); the Sino-UK Education Fund, UK (OP202006); and BBSRC, UK (RM32G0178B8).

**Data Availability Statement:** The CBIS-DDSM can be available on https://www.kaggle.com/datasets/awsaf49/cbis-ddsm-breast-cancer-image-dataset (accessed on 1 March 2023). The INbreast are be downloaded at https://www.kaggle.com/datasets/martholi/inbreast (accessed on 1 March 2023).

**Conflicts of Interest:** The authors declare no conflict of interest.

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
