# Peer review of "GFNet: A Deep Learning Framework for Breast Mass Detection"

_electronics, doi:10.3390/electronics12071583_

Round 1

Reviewer 1 Report

 Authors developed a novel mass detection framework named GFNet. The GFNet is comprised of three modules, including pre-processing, patch extraction, and mass detection. In the pre-processing module, the pectoral muscle is removed to obtain the breast-only image, which is enhanced by a classic image contrast enhancement algorithm. In the patch extraction module, mammograms' gradient field convergence features (GFCFs) are extracted and selected, based on which the breast mass patches can be extracted correspondingly. Then, the extracted patches are classified into breast background and mass candidates by deep learning models. We proposed a novel false positive reduction method that takes morphological and texture features as input, after which we applied non-maximum suppression for further false positive reduction. The research work reported is interesting in the community. Some suggestions are listed below to improve the manuscript's quality .

1. The manuscript's motivations should be further highlighted in the manuscript, e.g., what problems did the previous works exist? How to solve these problems? 

2. The research gaps in the abstract and introduction should be clearly expressed. Please rewrite this part.

3. The authors must clearly explain the difference(s) between the proposed method and similar works in the introduction.

4. The authors should further highlight the manuscript's innovations and contributions.

5. In section 3, authors should provide a flowchart for the proposed method.

6. In page 10, in Table 5, how to determine the vales of hyper-parameters? Please  explain it.

7.The literature review is poor in this paper. I hope that the authors can add some new references in order to improve the reviews. For example, https://doi.org/10.3389/fendo.2022.1057089;https://doi.org/10.1016/j.ins.2022.12.068; 10.1016/j.marstruc.2022.103338 and  so on.

8.There are some grammatical errors seen in the paper. Check carefully for a few clerical errors and formatting issues.

Reviewer 2 Report

This paper proposed a deep learning framework named GFNet for breast mass detection, which includes three modules, i.e., pre-processing, patch extraction, and mass detection. A feature extraction technique based on GFCF and a false positives reduction method were utilized. The topic is of interest to the medical image analysis community, the reported experimental results look promising, and the manuscript is well written in general, so I recommend to accept the paper after minor revisions. 

My suggestion to further improve the paper is that, since similar techniques used in the proposed scheme have appeared in existing publications, it could be helpful to highlight the potential advantages of the scheme by providing more discussions on comparison with related works in medical image analysis, especially those also used pre-processing, patch extraction, and deep learning based detection (e.g., "Adaptive Squeeze-and-Shrink Image Denoising for Improving Deep Detection of Cerebral Microbleeds" in MICCAI'21). Besides, please check and correct the typos in the paper, e.g., in line 511, was the variable n in latex format?

Author Response

  1. My suggestion to further improve the paper is that, since similar techniques used in the proposed scheme have appeared in existing publications, it could be helpful to highlight the potential advantages of the scheme by providing more discussions on comparison with related works in medical image analysis, especially those also used pre-processing, patch extraction, and deep learning based detection (e.g., "Adaptive Squeeze-and-Shrink Image Denoising for Improving Deep Detection of Cerebral Microbleeds" in MICCAI'21).

Response: Thank you for your comments. This paper is very interesting and are talked about in the Introduction.

  1. Besides, please check and correct the typos in the paper, e.g., in line 511, was the variable n in latex format?

Response: Thank you for your comments. We check and correct the typos in this paper.

‘With the increase of n, the accuracy seems to be saturated while the FPI increases, which indicates that  should be carefully chosen instead of setting it as large as possible.’

Round 2

Reviewer 1 Report

This paper can be accepted now.